# Association of Leopard Cat Occurrence with Environmental Factors in Chungnam Province, South Korea

**DOI:** 10.3390/ani13010122

**Published:** 2022-12-28

**Authors:** Ok-Sik Chung, Jong Koo Lee

**Affiliations:** 1Space and Environment Laboratory, Chungnam Institute, 73-26 Institute Road, Gongju 32589, Republic of Korea; 2Division of Life Sciences, College of Sciences and Bioengineering, Incheon National University, 119 Academy-ro, Yeonsu-gu, Incheon 22012, Republic of Korea

**Keywords:** species distribution modelling, leopard cat, geographic information system, spatial analysis

## Abstract

**Simple Summary:**

Understanding how environmental factors influence wildlife species is important for effective management. This study was conducted to address leopard cats’ distribution according to various environmental factors across Chungnam Province, South Korea, using two analytical approaches: classical statistical (i.e., logistic regression) and machine learning (i.e., boosted regression trees) methods. Results identified that higher leopard cat distribution was observed in the areas with lower elevation, closer to roads and water sources, and lower human population densities. The results also show that two methods can be used in a complementary manner for effective wildlife management.

**Abstract:**

This study was conducted to investigate the association of leopard cat (*Prionailurus bengalensis*) occurrences and environmental factors in Chungnam Province, South Korea, using two different analytical approaches for binomial responses: boosted regression trees and logistic regression. The extensive field survey data collected through the Chungnam Biotope Project were used to model construction and analysis. Five major influential factors identified by the boosted regression tree analysis were elevation, distance to road, distance to water channel/body, slope and population density. Logistic regression analysis indicated that distance to forest, population density, distance to water, and diameter class of the forest were the significant explanatory variables. The results showed that the leopard cats prefer the areas with higher accessibility of food resources (e.g., abundance and catchability) and avoid the areas adjacent to human-populated areas. The results also implied that boosted regression and logistic regression models could be used in a complementary manner for evaluating wildlife distribution and management.

## 1. Introduction

The leopard cat (*Prionailurus bengalensis*) is one of the top predator species and the only wild feline carnivore species left in South Korea [1,2,3]. The leopard cat was widely distributed across South Korea. However, due to the widespread application of rodenticide, habitat loss, excessive hunting and trapping, roadkill, and competition with feral cats and dogs, their population declined rapidly [2,4]. Although several studies of individual behaviour and habitat characteristics of leopard cats are available in South Korea (e.g., [1,5,6]), our understanding of managing population at the larger scale is still limited [2,7].

The Biotope Mapping Project in Chungnam Province of South Korea was conducted from 2008 to 2014 to support land planning and environmental policy decisions. As a result, it included the intensive wildlife data sampling campaign on a large scale in the area; over ~3500 wildlife sampling points across the province were surveyed through 3–4 phases [8]. Among them, approx. 1500 mammal sampling points were surveyed. These abundant field survey data can provide a timely opportunity to investigate the association between leopard cat suitability and other environmental factors.

Species distribution modelling (SDM) incorporates the species occurrence data and environmental, geographical, and anthropogenic information to predict an organism’s suitability [9]. Currently, a wide variety of statistical/analytical tools for SDM are implemented to evaluate species distribution depending on the nature of the data [10]. Boosted regression tree (BRT) modelling is a machine learning technique that has increased interest in species distribution modelling [11,12]. BRT modelling is a hybrid of machine learning and decision tree model (e.g., [13]). Instead of constructing a single best model, BRT modelling combines the “boosted” relatively simple tree models while optimizing its prediction performance [12]. The BRT model can accommodate different types of explanatory variables and handle their interaction automatically, and it is tolerant to missing values and outliers [11,12,14].

On the other hand, conventional logistic regression has been widely applied to analyze habitats and species distribution using binomial presence-absence data [15]. Logistic regression modelling is one type of generalized linear modelling method, which is broadly available in most statistical packages [16]. Whereas interpretation of the significant explanatory variables is straightforward [15], it often requires careful evaluation of model assumptions and sampling design [17], and it might have low data efficiency [16].

Therefore, the modelling approach for species distribution should be chosen by considering research goals, the nature of data, and each model’s usability [10]. However, relatively few comparative studies for species distribution and habitat suitability have been conducted [9,15], particularly for small mammals such as leopard cats [18,19]. The objective of this study is to evaluate the factors associated with the distribution of leopard cats using the BRT model and to compare the outcomes with the logistic regression model. The research findings would be beneficial for planning conservation and management for leopard cats.

## 2. Materials and Methods

### 2.1. Study Area

The study area was the Province of Chungcheongnam (“Chungcheongnam-do” or “Chungnam”, hereafter referred to as Chungnam), located in the mid-western part of South Korea (35°58′~37°03′ N, 125°31′~127°38′ E; Figure 1). The province has 8226 km^2^ of a geographic area as of 2017. The land of the province consists mainly of hilly or gentle lowlands. Approximately 66% of the total land area is less than 100 m above sea level and less than 5 degrees in slope [20]. More than 70% of total land cover is forests (50%) and agricultural lands (26%) [21]. The total forest area is 4080 km^2^, and most forests (about 80%) were young and have been reforested since the 1970s [20]. As of 2018, the province’s forests consist quite evenly of coniferous, deciduous broadleaf, and mixture forests, and the average stocking volume was 139 m^3^/ha [22].

A humid continental climate with hot and humid summers but dry winter is the representative climate of the Chungnam Province. The average monthly temperature ranges from −1.7 °C (in January) to 25.3 °C (in August), averaging 12.2 °C annually. Annual precipitation is 1310 mm, concentrated in summer, whereas winter has less than 10% of rainfall [23].

### 2.2. Data Collection, Processing, and Analyses

During the Biotope Mapping Project, entire land area of the Chungnam Province were classified into biotopes and mapped. A total of 1483 sampling points were randomly assigned (Figure 1) at the biotopes and were surveyed from 2005 to 2011. Because leopard cats usually mark their territory with feces, researchers thoroughly investigated the feces of leopard cats on the ground or rocks within a 50 m radius of each sampling point during winter season in which the understory plant would be relatively little.

Geographical and anthropogenic variables were extracted from digital thematic maps (Korea National Geographic Information Institute, Suwon, South Korea) such as digital elevation model, road network and water channel maps, and population map (Table 1). From the thematic maps, population density (POP_DEN), distances to the nearest road (DIST_RD) and water channel/body (DIST_WAT), slope (SLOPE) and elevation (ELEV) were calculated for each wildlife sampling point.

Forest attribute variables, including distance to the nearest forest (DIST_FOR), forest type (F_TYPE), diameter class (DIA_CL), age class (AGE_GL), tree density class (DEN_CL), forest area (F_AREA), were obtained from the Digital Forest Cover Type Map (Korea Forest Service, Daejeon, South Korea). The forest attribute variables were extracted only if the sampling points were located within a forest, and in this case, DIST_FOR was treated as 0. All spatial data processing was conducted using the ArcGIS Desktop (ver. 10.8.1 ESRI Inc. Redlands, CA, USA).

Boosted regression tree modelling was run following the guidance of Elith et al. [12]. The backward variable selection method was implemented for logistic regression analysis to find the most parsimonious model. The logistic regression and BRT models were fitted with the base and *gbm* packages [24] of R Statistical Software [25].

## 3. Results

The traces of leopard cats were observed from 374 sampling points out of 1483. (Figure 1). The BRT modelling result indicated that elevation (relative influence: 15.3%), distance to road (15.2%), distance to water (15.1%), slope (14.2%), and population density (13.5%) were prominent factors influencing leopard cat occurrence. Those five predictors contributed approximately 73% of the model prediction. Kappa and true skill statistics (TSS) were 0.863 and 0.807, respectively. Partial dependency plots indicated that the occurrence of leopard cats declined rapidly beyond ~300 m asl (Figure 2). In addition, the leopard cat occurrence decreased with increases in distance to the nearest water body and roads. Leopard cat occurrence was predicted to be intermediate in flat areas and relatively abundant in steep areas. However, a low leopard occurrence prediction was made in the gentle area (<~5% slope). Higher population density also tended to decrease the leopard cat occurrence.

The result of the logistic regression identified the distance to the nearest forest, population density, distance to water, and diameter class as the significant explanatory variables (Table 2). All the significant variables had negative associations with leopard cat occurrences. Among diameter class variables, only diameter class 1 (stands where more than 50% of crown projection area was occupied by seedlings and saplings < 6 cm dbh) was significant (*p* = 0.007).

Correlations among the significant explanatory variables were not significant except between elevation and slope (coefficient: 0.699; *p*-value: <0.001). Correlation coefficients among other variables ranged from 0.007 (elevation vs. population density; *p*-value: 0.789) to 0.216 (distance to road vs. distance to water; *p*-value: <0.001).

## 4. Discussion and Conclusions

Although some leopard cat studies in South Korea reported that higher elevation had a positive association with leopard cat occurrences (e.g., [1,26]), there are also observations of leopard cats’ preferences to lower-elevation areas (e.g., [27]). Those studies in South Korea may not be compatible since those were primarily conducted in somewhat disparate landscapes, such as Kangwon Province or a national park mainly located in remote and rugged mountainous areas with much higher elevations. In addition, Choi et al. [18] suggested habitat preferences can differ by the scale of the area; they reported that the leopard cat prefers meadows and paddy fields over forests in the landscape scale. Therefore, the observed rapid drop in leopard cat occurrence in high-elevation areas (Figure 2a) appears due to the geographical disadvantage (e.g., increased distance to forest edge or agricultural lands).

Roads and adjacent areas increase the visibility of prey, resulting in higher hunting success [28]. This suggestion may partially explain the observed negative association between leopard cat occurrence and the distance to roads. The preferences for ease of movement and visibility can also be found in other feline species such as bobcats (*Lynx rufus*; [29]), lions (*Panthera leo*; [30]) and cheetahs (*Acinonys jubatus*; [31]). In addition, roads can be used to forage for food, such as frogs depending on weather conditions [4]. Furthermore, their preference can also explain a relatively higher rate of leopard cat roadkill (e.g., [6,18]).

The preference of leopard cats for water sources is in line with other previous studies in South Korea (e.g., [1,5,19]). Preference for inland wetlands has been reported from a wide variety of wildlife species [6]. In a radio-tracking study of leopard cats, Choi et al. [27] reported that every individual’s location was concentrated around inland wetlands. Depending on season and location, small amphibians and reptiles can be important prey for the leopard cat [4,7,28]. In addition, the riparian area can provide abundant vegetation for refuge [6,26] and waterbirds for prey.

Low human population density is one of the most substantial spatial requirements for suitable feline species’ habitats [2]. Choi et al. [18] observed a clear avoiding behaviour of leopard cats against human residential areas and activities. In South Korea, feral cats’ population and spatial range are expanding rapidly, particularly from human residential areas [26]. Expansion of feral cat distribution to wild cat’s habitat can intensify the competition for prey and habitat resources [32]. The sudden drop of leopard cat’s occurrence in the area of gentle slopes (Figure 2d), where human residential areas are mainly located, might be due to the conflicts between human and feral cats or dogs.

Negative associations with the distance to forest and low leopard cat occurrence in the juvenile forest may imply that the leopard cat’s distribution depends on the abundance of food resources and habitat cover. Leopard cats prefer the forest edge since those areas are easily accessible forests for refuge and agricultural landscapes for prey [28]. Choi et al. [18] and Grassman et al. [33] suggested that the distribution of leopard cats should be closely related to the abundance of rodent species. Rodent species’ preference for higher vegetation cover [34] may result in higher leopard cat occurrence in those areas. In addition, thick forest cover is also important for rest and possible breeding [28,35]. Therefore, maintaining higher vegetation complexity is essential to manage the leopard cat population [35].

Although the initially hypothesized logistic regression model failed to detect some influential explanatory variables from BRT, the two approaches yielded consistent results. For example, if we treated the elevation variables and slope variables as specific indicator variable set instead of continuous variables (i.e., I_elve300_, where I_elve300_ = 1 if elevation larger than 300 m asl, 0 if not; I_slope1-4_, where I_slope1-4_ = 1 if slope is larger than 1% but less than 4%, 0 if not), then those indicator variables in the logistic regression models were significant (coefficients: −0.793, −0.523; *p* = 0.002, 0.013, for I_elve300_ and I_slope1-4_, respectively; data not shown). Therefore, the results show that the logistic regression model can fail to detect the significant association if the response changes drastically (e.g., nonlinear relationship, sharp discontinuity) within a narrow interval of the explanatory variable. If it is the case, setting appropriate cutoffs is challenging for researchers while building the research hypothesis or experimental design.

Ranganathan et al. [28] warned that putting inappropriate indicator variables and highly correlated variables in the model hinders the detection of the true association of other variables in the logistic regression models. Unfortunately, handling ecological datasets is seldom free from those issues due to our lacking understanding of the complex nature of environmental interactions. Therefore, BRT can be advantageous because tree structures in the model automatically handle the interactions among explanatory variables [12]. Moreover, BRT can be immune to outliers, the difference in scales among explanatory variables, and missing values [12].

However, we want to emphasize that our results are not intended to diminish the usefulness of the logistic regression approach. BRT also has limitations, such as lacking information for *p*-values for significant variables and degree of freedom [12]. Misunderstanding modelling techniques, including overfitting, can also be an issue [36]. Furthermore, there is still lacking understanding of the performances of each analytical approach by spatial scales. For example, Lee and Song [5] reported that an analytical technique can yield different results by the scale of the analytic unit. Furthermore, even using an identical dataset, the significant explanatory variables can have the opposite relationships with response variables by employing different analytical approaches (e.g., [19]). Therefore, the results of this study should be interpreted as those two analytical approaches can be used complementarily (e.g., identifying significant candidate explanatory variables and checking linear/non-linear relationships using BRT and calculating *p*-values and quantifying uncertainty using logistic regression) rather than competing with each other.

## Figures and Tables

**Figure 1 animals-13-00122-f001:**
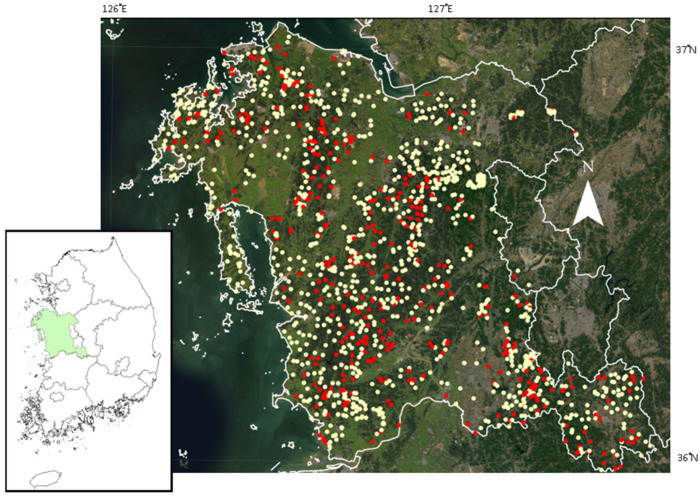
Locations of sampling points for leopard cats in the Chungnam Province, South Korea. Sampling points with and without the traces of leopard cats were marked with red and yellow, respectively.

**Figure 2 animals-13-00122-f002:**
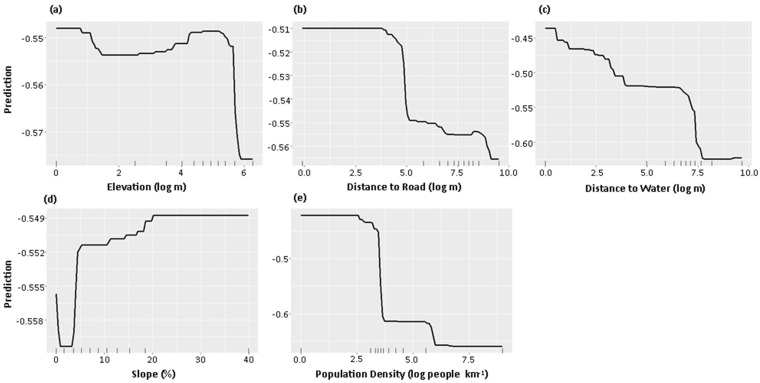
Partial dependence plots of a boosted regression tree analysis showing the relationships between the prediction of leopard cat occurrence and five major predictors. Those five predictors are: (**a**) elevation (relative influence: 15.3%) (**b**) distance to road (15.2%), (**c**) distance to water (15.1%), (**d**) slope (14.2%), (**e**) population density (13.5%), respectively.

**Table 1 animals-13-00122-t001:** Description of variables for model construction.

Variables	Notation	Unit	Note	Source ^†^
Leopard cat occurrence	-	-	binomial	CI
Population density	POP_DEN	people km^−2^	log-transformed	KNGII
Distance to road	DIST_RD	m	log-transformed	KNGII
Distance to water	DIST_WAT	m	log-transformed	KNGII
Distance to forest	DIST_FOR	m	log-transformed	KFS
Forest type	F_TYPE	-	categorical data	KFS
Diameter class	DIA_CL	-	categorical data	KFS
Age class	AGE_CL	-	categorical data	KFS
Density class	DEN_CL	-	categorical data	KFS
Elevation	ELEV	m	log-transformed	KNGII
Slope	SLOPE	%		KNGII
Forest area	F_AREA	m^2^	log-transformed	KFS
Perimeter: area ratio	PA_RATIO	m m^−2^		KFS

^†^ CI: Chungnam Institute; KFS: Korea Forest Service; KNGII: Korea National Geographic Information Institute.

**Table 2 animals-13-00122-t002:** Summary of logistic regression analysis. The notation of variables was described in Table 1 (SE: standard error).

Variable	Estimate	SE	z-Statistic	*p*-Value
Intercept	1.118	0.451	2.479	0.013
DIST_FOR	−0.113	0.064	−1.761	0.078
POP_DEN	−0.249	0.067	−3.728	<0.001
DIST_WAT	−0.140	0.04	−3.475	0.001
DIA_CL(1) ^†^	−0.835	0.312	−2.681	0.007
DIA_CL(2)	−0.052	0.268	−0.192	0.847
DIA_CL(3)	−1.177	0.8	−1.471	0.141

^†^ Diameter class 1: more than 50% of crown projection area was filled with less than 6 cm dbh trees; Diameter class 2: more than 50% of crown projection area was filled with ≥6 cm but <18 cm dbh trees; Diameter class 3: more than 50% of crown projection area was filled with ≥18 cm but <30 cm dbh trees.

## Data Availability

The data that support the findings of this study are available from the corresponding author upon reasonable request.

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
