# Peer review of "Association of Leopard Cat Occurrence with Environmental Factors in Chungnam Province, South Korea"

_animals, 2022, doi:10.3390/ani13010122_

Round 1

Reviewer 1 Report

I have read with great interest the manuscript of Chung and Lee with the title “Association of Leopard Cat Occurrence with Environmental Factors in Chungnam Province, South Korea”, and I found it to be of considerable validity both from the methodological point of view and for the important implications in the conservation of the leopard cat. In particular, the manuscript explores the effect of various factors on the distribution of leopard cats using the BRT model and compares the outcomes with the logistic regression model. The multimodel approach is certainly very useful for identifying the areas of the potential presence of species starting from presence-only data.

Besides, I feel that the emphasis should be placed on the results with ecological and conservation implications, not on the technical aspects concerning the performance of both models. I think this should be discussed in more detail in the text. Therefore, I would suggest restructuring the results and discussion parts, bringing forward the potential distribution of the leopard cats and the effects of the considered drivers revealed by the best models, and addressing the technical issues afterward and the justification for choosing both models.

I have five critical points that the authors should address:

1)    Reading the manuscript, the authors tend only to evaluate the factors associated with the distribution of leopard cats using two models and don't identify the areas predicted as suitable. Whereas in the context of species conservation, it is vital to assess their distribution and habitat use in order to prioritize the protection of species and their critical areas.

2)    It is important to choose the model with the best performance. Various research showed that boosted regression trees were not the top-performing models, compared with other models such as Ensembles, Random Forest, etc (e.g., Oppel et al. 2011).

3)    It would be necessary to have more details on the statistical analyses and the models used.

4)    The discussion might be too limited. Although interesting results are given, this may be of limited benefit when not linked to future leopard cat management aspects and the utilization of both models and their results for leopard cat conservation. Please add at the end of the discussion regarding "Habitat Conservation and Management Recommendations for Leopard cat" in relation to the main research findings

5)    Overall, the manuscript is easy to follow and concise, but there are some grammar issues (some of them identified below), so I recommend revising the English language. For example, in the section of “Simple Summary”: Line 10 (delete “the” before environmental), Line 11 (change "to effective" to "for effective") and (should be “cat” not “cat's”), Line 12 (should be “environmental” not “environment”) and (delete “different”), Line 14 (should be “cat” not “cat's”), etc.

In addition, I also have a series of minor comments:

Lines 14-16: Create the predictive habitat suitability or distribution map

Lines 27-29: How to evaluate the model you are using? We recommend that you use discriminant matrices such as the Area Under the Curve (AUC), Kappa, true skill statistics (TSS), Jaccard, and Sørensen

Lines 66-68: comparative studies related to what? Please add references after the phrase ”particularly for small mammals such as leopard cats”

Lines 68-70: I suggest adding an objective. This research is intended not only to evaluate environmental factors that affect the distribution of species but also to predict the suitability of habitat or distribution of leopard cats which is very beneficial for the conservation of this species.

Lines 79-80: Please provide the percentage for each land cover (forest and agricultural land)

Lines 97-98: “traces of leopard cats were recorded within a 50 m radius”, from what kind of survey activity? What survey method was used? Differences in survey techniques will bias the results 

Lines 99-111: Are all variables included in the model? What about the issue of multicollinearity between variables?

Lines 123-125: Present the percentage for each variable so that it quantitatively explains why these variables are considered to have the most influence on the distribution of leopard cat

Author Response

Dear Reviewer 1 

We deeply appreciate sincere comments on our manuscript. We tried to follow your comment as much as we can. We fully agree with your comments and it is very helpful to improve our manuscript. Please check this out. 

Thank you very much. 

Sincerely,

Jong Koo Lee

Reviewer 2 Report

Dear authors,

The manuscript entitled "Association of Leopard Cat Occurrence with Environmental Factors in Chungnam Province, South Korea" provides very interesting insights into the biology of this species and its conservation. I consider that the manuscript is well written and, given its simplicity, does not commit serious methodological errors.

However, I consider that more data should be provided in the Methods section on how the sampling was performed and it would be more clearly expressed what type of data was obtained to consider the presence/absence of the animal: tracks, hair, direct sightings, scat.... Also, why is the sampling not uniform throughout the territory? Although the data are very robust, it would be good to explain why there are some areas without sampling.

On the other hand, I would consider a basic issue to discuss the results: the intercorrelation of variables (higher altitude leads to less human presence, for example).

Author Response

Dear Reviewer 2

We deeply appreciate sincere comments on our manuscript. We tried to follow your comment as much as we can. We fully agree with your comments and it is very helpful to improve our manuscript. Please check this out. 

Thank you very much. 

Sincerely,

Jong Koo Lee
